# Clinical Manifestations and Outcomes of Tubercular Uveitis in Taiwan—A Ten-Year Multicenter Retrospective Study

**DOI:** 10.3390/medicina58030376

**Published:** 2022-03-03

**Authors:** Chun-Ju Lin, Ning-Yi Hsia, De-Kuang Hwang, Yih-Shiou Hwang, Yo-Chen Chang, Yueh-Chang Lee, Yung-Ray Hsu, Po-Ting Yeh, Chang-Ping Lin, Hsi-Fu Chen, Wei-Chun Jan, Wei-Yu Chiang, Ming-Ling Tsai

**Affiliations:** 1Department of Ophthalmology, China Medical University Hospital, China Medical University, Taichung 404, Taiwan; 2School of Medicine, College of Medicine, China Medical University, Taichung 404, Taiwan; 3Department of Optometry, Asia University, Taichung 413, Taiwan; 4Department of Ophthalmology, Taipei Veterans General Hospital, Taipei 112, Taiwan; m95gbk@gmail.com; 5School of Medicine, National Yang-Ming University, Taipei 112, Taiwan; 6Department of Ophthalmology, Chang Gung Memorial Hospital, Linkou 333, Taiwan; yihshiou.hwang@gmail.com; 7Department of Ophthalmology, Kaohsiung Medical University Chung-Ho Memorial Hospital, Kaohsiung 807, Taiwan; ycchang.oph@gmail.com; 8Department of Ophthalmology, Hualien Tzu Chi Hospital, Hualien 970, Taiwan; josephyclee@mail.harvard.edu; 9Department of Ophthalmology, Far Eastern Memorial Hospital, New Taipei 220, Taiwan; yungrayhsu@gmail.com; 10Department of Ophthalmology, National Taiwan University Hospital, Taipei 100, Taiwan; ptyeh67@gmail.com (P.-T.Y.); cpiling59@gmail.com (C.-P.L.); 11Department of Ophthalmology, Mackay Memorial Hospital, Hsinchu 300, Taiwan; aurora6th@yahoo.com.tw; 12Department of Ophthalmology, Mackay Memorial Hospital, Taipei 104, Taiwan; weijung.jan04@gmail.com; 13Department of Ophthalmology, Chang Gung Memorial Hospital, Kaohsiung 833, Taiwan; lcwarm021@gmail.com; 14Department of Ophthalmology, Taipei Tzu Chi Hospital, Tri-Service General Hospital, Taipei 231, Taiwan; doc30845@gmail.com

**Keywords:** antitubercular therapy, ethambutol, non-steroidal anti-inflammatory drugs, pyrazinamide, tubercular uveitis, vitrectomy

## Abstract

*Background and**Objectives*: This 10-year multicenter retrospective study reviewed the clinical manifestations, diagnostic tests, and treatment modalities of tubercular uveitis (TBU), including direct infection and indirect immune-mediated hypersensitivity to mycobacterial antigens in Taiwan. *Materials and Methods*: This retrospective chart review of patients with TBU was conducted at 11 centers from 1 January 2008 to 31 December 2017. We used a multiple regression model to analyze which factors influenced best-corrected visual acuity (BCVA) improvement. *Results*: A total of 79 eyes from 51 patients were included in the study. The mean age was 48.9 ± 16.4 years. The mean change of LogMAR BCVA at last visit was −0.21 ± 0.45. Diagnostic tools used include chest X-ray, chest computed tomography, Mantoux test, interferon gamma release test (QuantiFERON-TB Gold test), intraocular fluid tuberculosis polymerase chain reaction, and bronchial alveolar lavage. The clinical manifestations included 48% posterior uveitis and 37% panuveitis. In the sample, 55% of the cases were bilateral and 45% unilateral. There was 60.76% retinal vasculitis, 35.44% choroiditis, 21.52% serpiginous-like choroiditis, 17.72% vitreous hemorrhage, 12.66% posterior synechiae, 6.33% retinal detachment, and 3.80% choroidal granuloma. Treatment modalities included rifampicin, isoniazid, pyrazinamide, ethambutol, oral steroid, posterior triamcinolone, non-steroidal anti-inflammatory drugs, vitrectomy, and immunosuppressants. BCVA improved in 53.2% of eyes and remained stable in 32.9% of eyes. In the final model of multiple regression, worse initial BCVA, pyrazinamide, and receiving vitrectomy predicted better BCVA improvement. Ethambutol was associated with worse visual outcomes. Seven eyes experienced recurrence. *Conclusions*: This is the largest 10-year multicenter retrospective study of TBU in Taiwan to date, demonstrating the distribution of clinical manifestations and clinical associations with better treatment outcomes. The study provides a comprehensive description of TBU phenotypes in Taiwan and highlights considerations for the design of further prospective studies to reliably assess the role of ATT and vitrectomy in patients with TBU.

## 1. Introduction

Tuberculosis (TB) is an airborne disease mostly caused by Mycobacterium tuberculosis (MTB). It is a slowly progressive, chronic, necrotizing or non-necrotizing, and granulomatous or nongranulomatous infection, which usually infects the lungs. Only 10% of infected individuals become symptomatic. Moreover, 90% of those with latent TB remain infected without manifesting diseases. Alveolar macrophages acquire phagocytic and bactericidal functions and may limit the pulmonary infection. However, some organisms may escape through lymphatics or blood resulting in seeding of other organs, such as the cardiovascular system, gastrointestinal system, musculoskeletal system, genitourinary tract, central nervous system, skin, and eyes [1,2].

Ocular TB encompasses any impact by MTB in the eye, around the eye, or on its surface. It can be acquired by direct infection or an indirect immune-mediated hypersensitivity response to mycobacterial antigens, and is a great mimicker of various uveitis entities [3,4,5]. Primary ocular TB refers to conditions in which the eye is the initial port of entry, including conjunctival, corneal, and scleral diseases. Secondary TB refers to conditions in which the pathogen organisms spread to the eye hematogenously, including tubercular uveitis (TBU) [6].

There are two possible pathophysiological mechanisms of TBU: one is active spread via blood circulation and direct invasion of MTB into intraocular tissues, such as choroidal granuloma, and the other is delayed hypersensitivity immunological response to MTB perched other places in the body, not associated with local replication of the pathogen, like in serpiginous choroiditis. Therefore, combined anti-infectious and anti-inflammatory treatment are necessary.

The definite diagnosis of TBU is challenging due to the difficulty in demonstrating acid-fast bacilli on smears or histopathology in intraocular specimens. The paucity of available intraocular tissue is another issue. Molecular diagnostic techniques such as polymerase chain reaction (PCR) from aqueous or vitreous humor have low sensitivity due to the low bacterial load in ocular fluids and the thick cell wall of MTB [7,8,9,10,11]. For this reason, diagnoses of tubercular uveitis are mainly presumptive, based on ocular signs consistent with TBU, evidence of systemic TB, and a positive Mantoux test and/or interferon gamma release assay (IGRA).

This 10-year multicenter retrospective study reviews the clinical manifestations, diagnostic tests, and treatment modalities of TBU in Taiwan. We also use multiple regression model to analyze which factors will influence best-corrected visual acuity (BCVA) improvement.

## 2. Material and Methods

This retrospective chart review of patients with TBU was conducted at 11 Taiwan medical centers from 1 January 2008 to 31 December 2017 in compliance with the tenets of the Declaration of Helsinki. IRB approval was obtained for all sites. Standardization of uveitis nomenclature scoring of anterior chamber cell and flare; anterior and posterior synechiae; lens opacity or precipitates; and vitreous haze score (standardized Nussenblatt scheme) were recorded. The clinical manifestations of TBU in this study include anterior uveitis (granulomatous, nongranulomatous, broad-based posterior synechiae, iris nodules), intermediate uveitis (ciliary body tuberculoma, granulomatous, nongranulomatous with organizing exudates in the pars plana or peripheral area), posterior uveitis (choroidal granuloma, subretinal abscess, serpiginous-like choroiditis, retinal vasculitis, hemorrhagic retinitis, and choroiditis), panuveitis, neuroretinitis, and optic neuropathy. Rubeosis iridis, neovascular glaucoma, retinal detachment, and vitreous hemorrhage were also recorded.

The guidelines for diagnosis of TBU are as follows: (1) exclusion of other uveitis diagnoses; (2) a clinical history and signs compatible with TBU; (3) ocular examinations revealing direct evidence, positive acid fast bacilli, positive culture of MTB, or positive PCR of ocular fluid for MTB; (4) systemic investigations as corroborative evidence, including positive Mantoux test, positive IGRA (QuantiFERON-TB Gold test), positive chest images, or confirmed active extrapulmonary TB; and (5) positive therapeutic response to anti-TB therapy (ATT) [4,5,6,7,8,9,10,11,12,13,14,15,16,17].

The clinical manifestations, diagnostic tests, and treatment modalities of TBU were reviewed and recorded. Because this was a 10-year retrospective multicenter chart review study, the treatment strategy with regards to treating patients as active TB or latent TB could not be standardized across all centers. A multiple regression model was used to analyze which factors influenced BCVA improvement. As for the statistical methods, SAS 9.4 was used for analysis in this study. For comparison of cross-section data, one-way ANOVA was used for continuous data and Fisher’s exact test was used for categorical data. For comparison of serial data, principle of generalized linear mixed model (GLMM) was applied with use of GLIMMIX procedure in SAS.

To build a prediction model to find possible factors that might influence functional outcome (BCVA), multiple regression was performed with Proc Reg in SAS. The outcome was final change of LogMAR BCVA and the predictors (independent variables) included age, gender, location, initial BCVA, existence of co-morbidities, vitreous hemorrhage, posterior synechiae, choroidal granuloma, choroiditis, neovascular glaucoma, retinal detachment, retinal vasculitis, rubeosis iridis, serpiginous-like choroiditis, recurrence of uveitis and treatment modalities. The selection method was stepwise with entry significance level setting at 0.05 and stay significance level setting at 0.05.

## 3. Results

A total of 79 eyes from 51 patients were evaluated in this study. The study group consisted of 24 males and 27 females, and the mean age was 48.9 ± 16.4 years (range 10–82). The mean LogMAR BCVA at presentation was 0.57 ± 0.61. The mean LogMAR BCVA at last follow-up was 0.35 ± 0.58. The mean change of LogMAR BCVA at last visit was −0.21 ± 0.45. The mean follow-up period was 18.20 ± 24.05 months. Seven eyes experienced recurrence (Table 1).

The clinical manifestations comprised of 48% (38 eyes) posterior uveitis, 37% (29 eyes) panuveitis, and 15% (12 eyes) intermediate uveitis. In the sample, 55% cases were bilateral and 45% unilateral. There was 60.76% retinal vasculitis, 35.44% choroiditis (discrete choroidal lesions), 21.52% serpiginous-like choroiditis (multifocal lesions that develop in a serpiginoid pattern and merge), 17.72% vitreous hemorrhage, 12.66% posterior synechiae, 6.33% tractional retinal detachment, and 3.80% choroidal granuloma. No rubeosis irides or neovascular glaucoma was recorded (Figure 1). Vitreous hemorrhage and tractional retinal detachment could be complicated from retinal neovascularization but hard to confirm without the assistance of ultrawidefield fluorescein angiography. There was no recording of macular edema, papillitis, or choroidal neovascular complications.

The diagnostic tools utilized include chest X-ray, chest computed tomography (CT), Mantoux test, IGRA, intraocular fluid real-time PCR assay for MTB DNA, and bronchial alveolar lavage. The percentage of positive rate for individual diagnostic test were 3.8% bronchial alveolar lavage, 5.06% intraocular fluid MTB PCR, 7.59% Mantoux test, 27.85% chest CT, and 27.85% chest X-ray. The most sensitive test was IGRA with sensitivity up to 84.81%. The most frequent diagnostic patterns include 51% only IGRA positive; 11% both IGRA and chest CT positive; 9% IGRT, chest CT, and chest X-ray positive (Table 2).

The treatment modalities included 70.89% rifampicin, 68.35% isoniazid, 56.96% pyrazinamide, 50.63% ethambutol, 43.04% oral steroid, 19.83% posterior triamcinolone, 16.46% non-steroidal anti-inflammatory drugs (NSAIDs), 10.53% vitrectomy, and 4.11% immunosuppressants (Figure 2). Some authors performed MTB PCR on vitrectomy fluid but there were no detailed clinical data.

BCVA improved in 53.2% of eyes, remained stable in 32.9% of eyes, and worsened in 13.9% of eyes. The mean LogMAR BCVA significantly enhanced from 0.57 to 0.35 after treatment (Figure 3). We use a multiple regression analysis to see which factors influenced the degree of improvement of BCVA. The possible influencing factors consist of clinical conditions and treatment modalities. The clinical conditions include age, gender, location, initial BCVA, existence of co-morbidities, vitreous hemorrhage, posterior synechiae, choroidal granuloma, choroiditis, neovascular glaucoma, retinal detachment, retinal vasculitis, rubeosis iridis, serpiginous-like choroiditis, and recurrence of uveitis. In the final model of multiple regressions, worse initial BCVA, pyrazinamide, and receiving vitrectomy predicted better BCVA improvement. However, the use of ethambutol was associated with worse visual outcome after treatment. The R-square of the final model was 0.3248 (Table 3).

## 4. Discussion

In the literature, clinical manifestations of TBU include granulomatous or nongranulomatous acute anterior uveitis, broad-based posterior synechiae, iris nodules, ciliary body tuberculoma, granulomatous or nongranulomatous intermediate uveitis, posterior uveitis, panuveitis, neuroretinitis, and optic neuropathy [3,4,12,13]. The characteristics of posterior uveitis include choroidal tubercle, subretinal abscess, serpiginous-like posterior pole involvement, and periphlebitis of retinal vessels with accumulation of whitish material around retinal veins [14,15,16]. Hemorrhagic retinitis and focal choroiditis are usually located adjacent to the affected retinal veins [17].

This 10-year retrospective chart review study showed that the clinical manifestations of TBU in Taiwan contain retinal vasculitis, choroiditis, serpiginous-like choroiditis, vitreous hemorrhage, posterior synechiae, tractional retinal detachment, and choroidal granuloma. Although anterior segment manifestations could occur simultaneously, posterior segment involvement is the more common form of TBU in Taiwan. Retinal vasculitis is the most frequent form of posterior involvement, along with choroiditis and vitreous hemorrhage.

In Asian populations, retinal vasculitis is a common presentation of TBU; however, in a study of a referral eye center in Iraq, vitritis was a universal finding, while multifocal choroiditis was the most common fundus lesions (104 eyes; 82.5%) [18]. In some nonendemic countries (the United Kingdom and Netherlands), serpiginous-like choroiditis is as common as retinal vasculitis as a manifestation of TBU [19,20,21].

Most diagnoses of TBU in our study were based on ocular signs consistent with TBU and positive IGRA. The majority of cases with TBU did not have any history or symptoms of other systemic TB, and the majority of patients (72.1%) had a negative finding in their chest X-ray.

The Mantoux test is one of the few investigations from the 19th century [22]. A positive Mantoux test might indicate mycobacterial infection, regardless of whether or not there is clinical manifestation of the disease. In Taiwan, the Bacillus Calmette–Guérin (BCG) vaccination has been given routinely at birth since 1965 and is repeated in elementary school if one’s Mantoux test is negative. According to the data provided by a recent national survey in Taiwan, the BCG vaccination percentage is 97% among first grade students. Because of Taiwan’s current policy of BCG vaccination, Mantoux tests would show false-positive results. False-negative Mantoux tests in a patient with definite pulmonary and ocular tuberculosis have also been reported [23]. Therefore, Mantoux tests have seldom been used in the diagnosis of TBU in Taiwan.

Real-time polymerase chain reaction (PCR) assay has been applied for the detection of MTB complex in clinical specimens. The inherent difficulties in obtaining adequate intraocular histopathological samples for TB PCR limit its use [7,8,9,10,11]. Nevertheless, the collaborative ocular tuberculosis study (COTS)-1 study suggests that positive/negative intraocular fluid TB PCR reports may not affect management or treatment outcomes in a real-world scenario [24]. IGRA measures the interferon gamma production by T cells which are triggered by ESAT-6, CFP-10, and TB7.7 antigens [25]. It has been recommended to initiate early ATT in patients with positive IGRA results and high suspicion of TBU.

The methods used for TBU diagnosis showed great variety of sensitivity in Taiwan. The leading positive rate for individual diagnostic test was 27.85% chest CT, 27.85% chest X-ray, and 84.81% IGRA. We further analyzed the patterns of positive diagnostic tests for individual eyes. The most frequent patterns were 51% only IGRA positive followed by 11% IGRA and chest CT positive, and 9% IGRT, chest CT, and chest X-ray positive. Therefore, the majority of TBU diagnoses in the study was based on clinical manifestations and positive IGRA.

Tuberculous retinal vasculitis (TRV) is typically an obliterative periphlebitis, resulting in retinal nonperfusion, which may induce proliferative retinopathy. Long-term complications of TBU with peripheral neovascularization and recurrent vitreous hemorrhage may lead to retinal detachment and/or macular pucker [26]. Close monitoring for retinal neovascularization development, subsequent vitreous hemorrhage, and tractional retinal detachment are very important. Timely laser treatment and vitrectomy are essential. Good anatomical and visual improvement may be achieved with appropriate vitrectomy [27,28,29]. This was also demonstrated in the final model of multiple regression in our present study.

When investigating patients with uveitis, it is not uncommon for there to be no identifiable systemic disease, with the only positive test being an IGRA [25,30]. In such patients, the question arises as to whether the uveitis is related to latent TB or not and whether ATT would be beneficial. Recent studies suggest that in patients with vision-threatening uveitis with no identifiable cause who have latent TB, the recurrence rate of uveitis is noticeably lessened with concomitant TB and uveitis treatment [30,31].

The COTS consensus statements for the initiation of ATT in TBU is treatment with fixed dose combination anti-tubercular treatment (Rifafour e-275), which includes rifampicin, isoniazid, pyrazinamide, and ethambutol hydrochloride for the first 2 months [32]. However, the management of ATT for TBU poses a challenge because of differing opinions among uveitis specialists and infectious disease specialists. In Taiwan, the infectious disease specialists do not usually start ATT among patients only with positive immunologic tests (Mantoux test or IGRA) if these patients do not have active pulmonary TB. Therefore, some uveitis specialists prescribe ATT for the management of TBU on their own. Considering the possibility of ethambutol-related ocular complications, some uveitis specialists may avoid ethambutol use in patients with existing ophthalmic disease.

In our present study, about 70% of patients received rifampicin and isoniazid. Around half of the patients received pyrazinamide and ethambutol. Steroid, immunosuppressants [29], and NSAIDs were used as supplementary therapy. NSAIDs have been reported as a potential safe, simple, and cheap adjunction in the treatment of TB [33].

The mean LogMAR BCVA significantly increased from 0.57 to 0.35 after treatment. Therefore, we use multiple regression analysis to see which factors will influence the degree of improvement of BCVA. In the final model of multiple regressions, worse initial BCVA, pyrazinamide, and receiving vitrectomy predicted better BCVA improvement.

Pyrazinamide is bactericidal for MTB. It is largely absorbed from the gastrointestinal tract and penetrates most tissues, including cerebrospinal fluid (CSF). Pyrazinamide and isoniazid are fairly lipophilic small molecules and therefore optimal drugs for the management of central nervous system (CNS) infections [34]. It appears to accelerate the sterilizing effect of isoniazid and rifampin. Unless the bacteria are resistant to these agents, isoniazid and pyrazinamide must be included in the treatment of tuberculous meningitis [35].

We also found use of ethambutol to be associated with worse visual outcome. Ethambutol is a bacteriostatic drug and has significant ocular toxicity, especially in patients with pre-existing ophthalmic disease, such as TBU. Optic neuritis is the most common complication and could happen in axial and periaxial forms. Axial optical neuritis manifests with decreased central visual acuity and green color perception and is associated with macular degeneration. Periaxial optic neuritis results in paracentral scotomas with normal vision and color perception. Other ocular side effects consist of extraocular muscle paresis, photophobia, and toxic amblyopia [6,36].

Although it would be statistically easier to manage single eye data from single patient, using eye-level data or individual-level data for ophthalmology study has been a long-standing debate. A very thorough review by Murdoch mentions many shortcomings and possible biases that might come from single-eye-per-individual data: (1) Unless we are certain that the disease has absolutely no laterality, choosing single eye has the risk of potential bias. (2) The major disadvantage of this approach is the loss of information. As the author described, the data might be statistically valid, but the power and precision of the analysis are less than optimal. (3) Another disadvantage of this approach is the potential for biases arising through the choice of which data to use [37].

Often, if data are only available on a single eye in an individual because of incomplete data or for other reasons, that single eye is included in the analysis. Bias could occur if some individuals have data on both eyes, only one of which is “randomly” selected for analysis, while other individuals have data on only one eye which is automatically included (that is, not randomly selected). The non-random selection of eyes can introduce bias. Similar caveats apply to the choice of the first eye with disease, worse/better eye, or operated eye.

They also mentioned some disadvantages of pooling or averaging results from right and left eyes. Meanwhile, in their review, there were up to 20% of qualitative studies which used data from all eyes (eye-level data). Because pros and cons exist for both types of data utilization, the author provided a formula to calculate the intraclass correlation coefficient and it is identical to kappa statistic. They suggested if this indicator is less than 0.3, the correlation between eyes can be treated as acceptable. The formula is: (P2eyes-Peye^2)/(Peye (1-Peye)). Where P2eyes is the proportion of individuals with the finding in both eyes and Peye is the proportion of eyes with the finding. The result of this coefficient from our data is −0.29.

Therefore, we opine that it is acceptable to use eye-level data for analysis in our study.

The 1-year follow-up of the COTS-1 of TRV showed no significant therapeutic effects of ATT [12]. Nevertheless, this 10-year multicenter retrospective study in Taiwan found that pyrazinamide for the treatment of TBU could obtain better BCVA improvement.

There are some limitations of this study. First, some diseased eyes came from the same patients. This might induce some bias for clinical manifestations that could be influenced by personal characteristics. However, because of the very low incidence of this disease entity, we opine that only including a single diseased eye from each person might also possibly exclude many valuable information that might be person-independent and also weaken the power of this study by further decreasing sample size. We treated our study as an innovative but preliminary one, which will inspire more stringent large-scaled intervention studies in the future. The other limitations of this study are the retrospective methodology, the lack of control group analysis, and the lack of standardized treatment protocols, which led to missing data and inability to account for severity of inflammation in outcomes analyses. However, this is the largest data set of TBU to date in Taiwan using 10-year data collected from 11 tertiary referral ophthalmology centers. The current study addresses the multicenter data and standardized criteria for inclusion and treatment outcomes.

Based on this study of TBU in Taiwan, IGRA is a helpful diagnostic tool in uveitis patients and it should be covered by Taiwan national health insurance reimbursement in the future. The presence of active typical ocular inflammation and a positive IGRA, with or without other systemic signs, could be considered as possible TBU.

In this long-term multicenter retrospective study, BCVA showed significant improvement after treatment. The final model of multiple regression showed that worse initial BCVA, pyrazinamide therapy, and receiving vitrectomy predicted better BCVA improvement. Use of ethambutol was associated with worse visual outcome after treatment. The current study provides a comprehensive description of TBU phenotypes in Taiwan and highlights considerations for the design of further prospective studies to reliably assess the role of ATT and vitrectomy in patients with TBU.

## Figures and Tables

**Figure 1 medicina-58-00376-f001:**
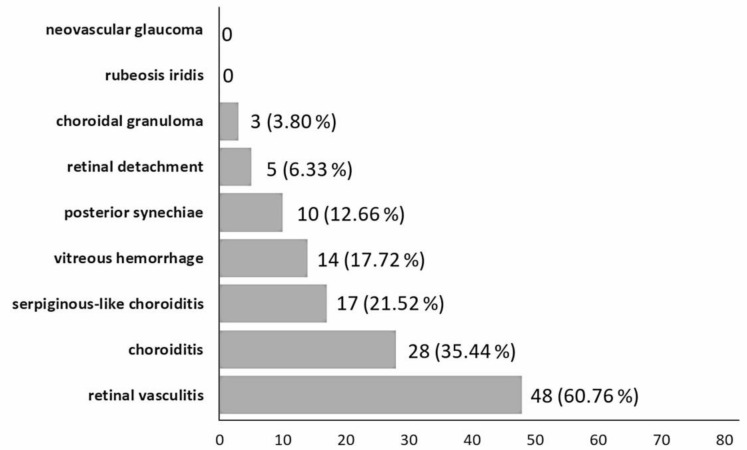
Breakdown of clinical manifestations of tubercular uveitis.

**Figure 2 medicina-58-00376-f002:**
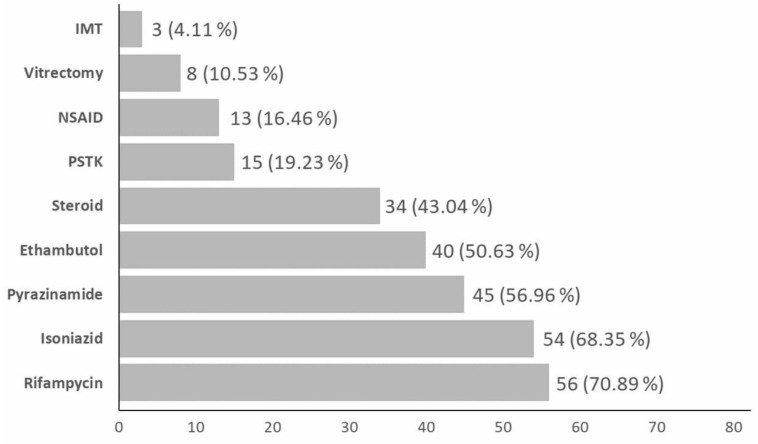
Breakdown of treatment modalities. IMT: immunosuppressants, PSTK: posterior subtenon kenalog, NSAID: non-steroidal anti-inflammatory drug.

**Figure 3 medicina-58-00376-f003:**
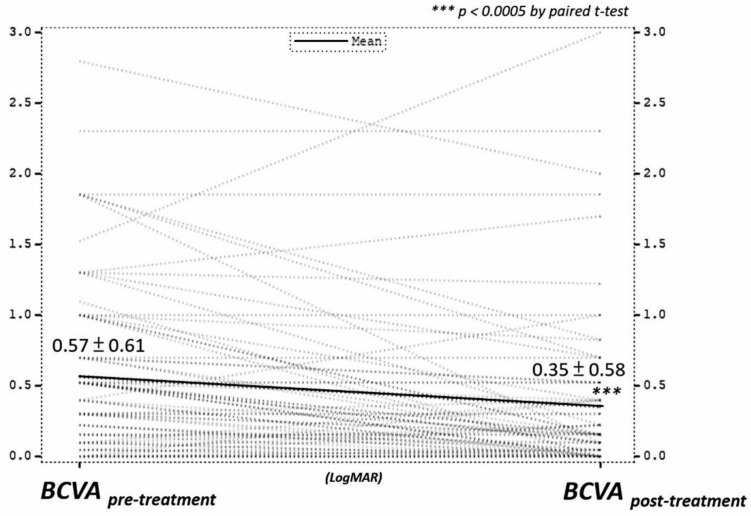
The significant BCVA improvement after treatment. *** *p* < 0.0005 by paired *t*-test.

**Table 1 medicina-58-00376-t001:** The main clinical characteristics.

No. of Eyes	79 (51 Patients)
Male/female (% of male)	24/27 (47.06%)
Mean age (years ± SD)	48.9 ± 16.4
Mean BCVA at presentation (LogMAR)	0.57 ± 0.61
Mean BCVA at last F/U (LogMAR)	0.35 ± 0.58
Mean change of BCVA (LogMAR)	−0.21 ± 0.45
No. of recurrence (%)	7 (8.86%)
Follow-up period (months)	18.20 ± 24.05

**Table 2 medicina-58-00376-t002:** Patterns of positive diagnostic tests among patients.

Diagnostic Tests						No. of Positive Test	No. of Eyes	%
MT	BAL	IFPCR	CXR	CCT	IGRA			
	+	+	+	+	+	5	2	3%
			+	+	+	3	7	9%
	+		+	+		3	1	1%
+					+	2	1	1%
		+			+	2	2	3%
			+		+	2	4	5%
				+	+	2	9	11%
+					+	2	2	3%
+			+			2	1	1%
			+	+		2	2	3%
					+	1	40	51%
				+		1	1	1%
			+			1	5	6%
+						1	2	3%
							79	100%

MT: Mantoux test, BAL: bronchial alveolar lavage, IFPCR: intraocular fluid TB PCR, CXR: chest X-ray, CCT: chest CT, IGRA: interferon gamma release assay.

**Table 3 medicina-58-00376-t003:** In the final model of multiple regression, worse initial BCVA (ValmFirst), treatment of pyrazinamide (TxPyra), and receiving vitrectomy (TxVitr) predicted better BCVA improvement. Use of ethambutol (TxEtha) was associated with worse visual outcome after treatment. R^2^ of final model: 0.3248.

Analysis of Variance					
Source	DF	Sum of Squares	Mean Square	F Value	Pr > F
Model	4	4.02949	1.00737	6.86	0.0001
Error	57	8.37522	0.14693		
Corrected Total	61	12.40471			
Variable	Parameter Estimate	Standard Error	Type II SS	F Value	Pr > F
Intercept	0.05795	0.09698	0.05246	0.36	0.5525
ValmFirst	−0.27612	0.08888	1.418	9.65	0.0029
TxEtha	0.2987	0.10758	1.13278	7.71	0.0074
TxPyra	−0.40127	0.11489	1.79237	12.2	0.0009
TxVitr	−0.39099	0.18499	0.65637	4.47	0.0389
			Model R^2^	0.3248	

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
