# Peer review of "Clinical Manifestations and Outcomes of Tubercular Uveitis in Taiwan—A Ten-Year Multicenter Retrospective Study"

_medicina, 2022, doi:10.3390/medicina58030376_

Round 1
Reviewer 1 Report
The major issue with the study is the statistical analysis. Authors included both eyes of some of the patients which is not acceptable as it violates the assumption of independence.
Moreover, the authors need to present in detail the results of their multivariate analysis.
Author Response
Thank you for your precious opinions.
- We can’t agree with you more that the inclusion of both eyes of the same patient into the study will bring some bias. This might induce some bias for clinical manifestations that could be influenced by personal characteristics. But because of the very low incidence of this disease entity, as you can see only less than 100 cases during 10-year follow-up period, we opine that only including single diseased eye from each person might also possibly exclude many valuable information which might be person-independent and also weaken the power of this study by further decreasing sample size. Basically we treated our study as an innovative but preliminary one, which will inspire more stringent large-scaled intervention studies in the future. Because this is an important point must be mentioned, we add some lines in the section of study limitation section to address this issue after your precious suggestion.
Please see page 17, para 3, line 1-8.
- About the multivariate analysis, the presentation of the analysis process in detail is necessary as you suggested. Therefore, we added more crucial information about it in the statistical section, including all variables involved and also the selection method we used to build the prediction model.
To build a prediction model to find possible factors that might influence functional outcome (BCVA), multiple regression was performed with Proc Reg in SAS. The outcome was final change of LogMAR BCVA and the predictors (independent variables) included age, gender, location, initial BCVA, existence of co-morbidities, vitreous hemorrhage, posterior synechiae, choroidal granuloma, choroiditis, neovascular glaucoma, retinal detachment, retinal vasculitis, rubeosis iridis, serpiginous-like choroiditis, recurrence of uveitis and treatment modalities. The selection method was stepwise with entry significance level setting at 0.05 and stay significance level setting at 0.05.
Please see page 9, para 2, line 1-9.
Reviewer 2 Report
Uveitis caused by MTB is a disease that can be a diagnostic and therapeutic challenge. Pathogens change and with them the diseases they cause, so it is so important to know the specificity of the inflammation caused by a given bacterium.
In uveitis, the frequency of the individual eye lesions is one of the most important guidelines when making initial diagnosis and treatment. The Authors rightly included these data in the abstract.
The discussion largely includes data from the Authors' research. I suggest including more data from outside of Taiwan, because the essence of the discussion is to analyze your own results against those from other researchers.
Authors concluded that the majority of tubercular uveitis diagnoses in the study was based on clinical manifestations and positive IGRA. It seems that it is currently the dominant method of TBU diagnostics in the world. However, it is worth mentioning that IGRA tests are relatively often false-positive. Therefore, the quantitative rather than qualitative assessment in IGRA tests is considered to be of particular value. Do the authors have information about the type of test and its quantitative results in relation to the laboratory standard?
The TBU abbreviation is not expanded on its first use - it should be corrected.
Do the Authors have any information on the choice of a specific treatment regimen in specific patients? One of the most important conclusions in the study is that pyrazinamide is more effective and ethambutol less effective. Due to the specificity of these drugs, this can be considered very likely. But due to the potential bias associated with the non-random selection of a specific treatment regimen, this conclusion should be expressed with caution.
The article is well written, contains relevant data, and can be expected to be cited for many years to come. I suggest slightly changing the discussion and publishing.
Author Response
Reviewer 2
Uveitis caused by MTB is a disease that can be a diagnostic and therapeutic challenge. Pathogens change and with them the diseases they cause, so it is so important to know the specificity of the inflammation caused by a given bacterium.
In uveitis, the frequency of the individual eye lesions is one of the most important guidelines when making initial diagnosis and treatment. The authors rightly included these data in the abstract.
The discussion largely includes data from the Authors' research. I suggest including more data from outside of Taiwan, because the essence of the discussion is to analyze your own results against those from other researchers.
A: Thank you so much for your comment. We’ve included more data from outside of Taiwan.
In Asian populations, retinal vasculitis is a common presentation of TBU; however, in a study of a referral eye center in Iraq, vitritis was a universal finding, while multifocal choroiditis was the most common fundus lesions (104 eyes; 82.5%).18 In some nonendemic countries (the United Kingdom and Netherlands), serpiginous-like choroiditis, is as common as retinal vasculitis as a manifestation of TBU.19-21
Please see page 12, para 3, line 1-3 and page 13, para 1, line 1-2.
Please see reference 18-21.
Authors concluded that the majority of tubercular uveitis diagnoses in the study was based on clinical manifestations and positive IGRA. It seems that it is currently the dominant method of TBU diagnostics in the world. However, it is worth mentioning that IGRA tests are relatively often false-positive. Therefore, the quantitative rather than qualitative assessment in IGRA tests is considered to be of particular value. Do the authors have information about the type of test and its quantitative results in relation to the laboratory standard?
A: Thank you so much for your comment. QuantiFERON-TB Gold test has been mostly used in Taiwan with quantitative levels. However, it’s a pity that the quantitative results were not available from the data provided from different hospitals. Nevertheless, the diagnosis of TBU include a clinical history and signs compatible with TBU and exclusion of other uveitis diagnoses, not only based on positive IGRA results.
Please see page 8, para 2, line 5.
The TBU abbreviation is not expanded on its first use - it should be corrected.
A: Thank you so much for your reminding. We’ve expanded TBU abbreviation on its first use. Please see page 6, para 2, line 7.
Do the Authors have any information on the choice of a specific treatment regimen in specific patients? One of the most important conclusions in the study is that pyrazinamide is more effective and ethambutol less effective. Due to the specificity of these drugs, this can be considered very likely. But due to the potential bias associated with the non-random selection of a specific treatment regimen, this conclusion should be expressed with caution.
A: Thank you so much for your comment. We’ve changed the description.
The COTS consensus statements for the initiation of ATT in TBU is treatment with fixed dose combination anti-tubercular treatment (Rifafour e-275), which includes rifampicin, isoniazid, pyrazinamide, and ethambutol hydrochloride for the first 2 months.33 However, the management of ATT for TBU poses a challenge because of differing opinions among uveitis specialists and infectious disease specialists. In Taiwan, the infectious disease specialists do not usually start ATT among patients only with positive immunologic tests (Mantoux test or IGRA) if these patients do not have active pulmonary TB. Therefore, some uveitis specialists prescribe ATT for the management of TBU on their own. Considering the possibility of ethambutol-related ocular complications, some uveitis specialists may avoid ethambutol use in patients with existing ophthalmic disease.
Please see page 15, para 3, line 1-10 and page 16, para 1, line 1.
Please see reference 33.
There are some limitations of this study. First, some diseased eyes came from the same patients. This might induce some bias for clinical manifestations that could be influenced by personal characteristics. But because of the very low incidence of this disease entity, we opine that only including a single diseased eye from each person might also possibly exclude many valuable information that might be person-independent and also weaken the power of this study by further decreasing sample size. We treated our study as an innovative but preliminary one, which will inspire more stringent large-scaled intervention studies in the future.
Please see page 17, para 3, line 1-8.
The article is well written, contains relevant data, and can be expected to be cited for many years to come. I suggest slightly changing the discussion and publishing.
A: Thank you so much for your comment. We’ve improved the discussion.
In Asian populations, retinal vasculitis is a common presentation of TBU; however, in a study of a referral eye center in Iraq, vitritis was a universal finding, while multifocal choroiditis was the most common fundus lesions (104 eyes; 82.5%).18 In some nonendemic countries (the United Kingdom and Netherlands), serpiginous-like choroiditis, is as common as retinal vasculitis as a manifestation of TBU.19-21
Please see page 12, para 3, line 1-3 and page 13, para 1, line 1-2.
Please see reference 18-21.
The COTS consensus statements for the initiation of ATT in TBU is treatment with fixed dose combination anti-tubercular treatment (Rifafour e-275), which includes rifampicin, isoniazid, pyrazinamide, and ethambutol hydrochloride for the first 2 months.33 However, the management of ATT for TBU poses a challenge because of differing opinions among uveitis specialists and infectious disease specialists. In Taiwan, the infectious disease specialists do not usually start ATT among patients only with positive immunologic tests (Mantoux test or IGRA) if these patients do not have active pulmonary TB. Therefore, some uveitis specialists prescribe ATT for the management of TBU on their own. Considering the possibility of ethambutol-related ocular complications, some uveitis specialists may avoid ethambutol use in patients with existing ophthalmic disease.
Please see page 15, para 3, line 1-10 and page 16, para 1, line 1.
Please see reference 33.
There are some limitations of this study. First, some diseased eyes came from the same patients. This might induce some bias for clinical manifestations that could be influenced by personal characteristics. But because of the very low incidence of this disease entity, we opine that only including a single diseased eye from each person might also possibly exclude many valuable information that might be person-independent and also weaken the power of this study by further decreasing sample size. We treated our study as an innovative but preliminary one, which will inspire more stringent large-scaled intervention studies in the future.
Please see page 17, para 3, line 1-8.
Round 2
Reviewer 1 Report
The authors have now presented in detail the results of the regression analysis. However by including both eyes of the some patients, they violate the assumption of independence and this can lead to wrong results.
I understand that including only one eye of each patient will reduce the sample size but this will increase the quality of the manuscript. Alternatively you can take the mean of both eyes of the patients and include it as one observation but I would personally include just one eye (randomly selected or the first eye which presented the uveitis).
Best wishes
Author Response
Thank you for your comments.
Although it would be statistically easier to manage single eye data from single patient, using eye-level data or individual-level data for ophthalmology study has been a long-standing debate. A very thorough review by Murdoch38 mentions many shortcomings and possible biases that might come from single-eye-per-individual data:
- Unless we are certain that the disease has absolutely no laterality, choosing single eye has the risk of potential bias.
- The major disadvantage of this approach is the loss of information. As the author described, the data might be statistically valid, but the power and precision of the analysis are less than optimal.
- Another disadvantage of this approach is the potential for biases arising through the choice of which data to use. Often, if data are only available on a single eye in an individual because of incomplete data or for other reasons, that single eye is included in the analysis. Bias could occur if some individuals have data on both eyes, only one of which is “randomly” selected for analysis, while other individuals have data on only one eye which is automatically included (that is, not randomly selected). The non-random selection of eyes can introduce bias. Similar caveats apply to the choice of the first eye with disease, worse/better eye, or operated eye.
They also mentioned some disadvantages of pooling or averaging results from right and left eyes. Meanwhile, in their review, there were up to 20% of qualitative studies which used data from all eyes (eye-level data).
Because pros and cons exist for both types of data utilization, the author provided a formula to calculate the intraclass correlation coefficient and it is identical to kappa statistic. They suggested if this indicator is less than 0.3, the correlation between eyes can be treated as acceptable. The formula is: (P2eyes-Peye^2)/ (Peye (1-Peye))
Where P2eyes is the proportion of individuals with the finding in both eyes and Peye is the proportion of eyes with the finding. The result of this coefficient from our data is -0.29. Therefore, we opine that it is acceptable to use eye-level data for analysis in our study.
Please see page 17, para 3 and 4; page 18, para 1-3.
Please see reference 38.
- Murdoch IE, Morris SS, Cousens SN. People and eyes: statistical approaches in ophthalmology. Br J Ophthalmol. 1998; 82(8):971-3.
